# Diagnostic Value of the Vestibular Autorotation Test in Menière’s Disease, Vestibular Migraine and Menière’s Disease with Migraine

**DOI:** 10.3390/brainsci12111432

**Published:** 2022-10-25

**Authors:** Dan Liu, Jun Wang, E Tian, Zhao-qi Guo, Jing-yu Chen, Wei-jia Kong, Su-lin Zhang

**Affiliations:** 1Department of Otorhinolaryngology, Union Hospital, Tongji Medical College, Huazhong University of Science and Technology, Wuhan 430022, China; 2Institute of Otorhinolaryngology, Union Hospital, Tongji Medical College, Huazhong University of Science and Technology, Wuhan 430022, China; 3Key Laboratory of Neurological Disorders of Education Ministry, Tongji Medical College, Huazhong University of Science and Technology, Wuhan 430022, China

**Keywords:** vestibular autorotation test, Menière’s disease, vestibular migraine, differential diagnosis, horizontal gain

## Abstract

(1) Background: Vestibular migraine (VM) and Menière’s disease (MD) share multiple features in terms of clinical presentations and auditory-vestibular functions, and, therefore, more accurate diagnostic tools to distinguish between the two disorders are needed. (2) Methods: The study was of retrospective design and examined the data of 69 MD patients, 79 VM patients and 72 MD with migraine patients. Five vestibular autorotation test (VAT) parameters, i.e., horizontal gain/phase, vertical gain/phase and asymmetry were subjected to logistic regression. The receiver operating characteristic (ROC) curves were generated to determine the accuracy of the different parameters in the differential diagnosis of MD and VM. (3) Results: Our results showed that the horizontal gain of VAT significantly outperformed other parameters in distinguishing MD and VM. In addition, the sensitivity, specificity and accuracy of the horizontal gain were 95.7%, 50.6% and 71.6%, respectively, for the differentiation between VM and MD. In most MD patients, the horizontal gain decreased in the range of 3–4 Hz, while in most VM patients, horizontal gain increased in the range between 2–3 Hz. More MD with migraine patients had an increased horizontal gain when the frequency was less than 5.0 Hz and had a decreased horizontal gain when the frequency was greater than 5.0 Hz. (4) Conclusion: Our study suggested the VAT, especially the horizontal gain, as an indicator, may serve as a sensitive and objective indicator that helps distinguish between MD and VM. Moreover, VAT, due to its non-invasive and all-frequency nature, might be an important part of a test battery.

## 1. Introduction

Menière’s disease (MD) represents an idiopathic inner ear disease with spontaneous episodic vertigo, fluctuating sensorineural hearing loss, tinnitus and aural fullness [1]. MD is pathophysiologically associated with endolymphatic hydrops in the inner ear [2,3]. Vestibular migraine (VM) is a common disease characterized by episodic vertigo and/or disequilibrium in patients with current or previous migraine history [4,5]. At present, most of the assumptions about VM are premised on the understanding of migraine per se. However, it is common for VM and MD to present concomitantly. Neff et al. [6] reported that 23% of patients with VM had coexistenting MD and 28% of MD patients had symptoms of VM. Patients with MD may show migraine symptoms even during the attacks of vertigo [7], suggesting that MD and VM might pathologically share some common mechanisms [8,9]. Clinically, if MD is mistakenly diagnosed as VM, the ensuing treatment may fail to address the vestibular dysfunction deficit. On the other hand, if VM is misdiagnosed as end-stage MD, the patient might be improperly managed, surgically. Accurate differentiation between these two disorders could avoid mismanagement and lead to more effective treatment. Therefore, it is of great significance to find an effective indicator for differentiating the two diseases.

To date, the diagnosis of MD and VM is primarily based on the patient’s history and symptoms [1,4]. Despite advances in the classification of vestibular disorders [9], the differential diagnosis between MD and VM is often difficult in clinical practice because of the episodic and fluctuating nature of the two conditions and the considerable overlap of symptoms and eventually disease course. For example, so far, caloric testing is the most widely used laboratory test for assessing peripheral vestibulopathy [10,11]. Nevertheless, it is non-physiological and time-consuming and is characterized by a very low-frequency response (≈0.003 Hz), which is well below the frequency of the head movement during functional activities [12]. Additionally, the head impulse test, which now has been upgraded to video head impulse test (vHIT), can examine the high-frequency vestibulo–ocular reflex (VOR) for each semicircular canal with high accelerations impulses (4–7 Hz) [13,14,15]. However, vHIT relies upon the examiner to perform a reliable rapid head movement, and vHIT is not suitable for patients with a neck problem that restricts passive head movement [16,17]. The vestibular autorotation test (VAT) was first proposed by O’Leary and Davis in 1988 to provide a cost-effective solution for the assessment of vestibular function over a broad range of head movement frequencies during functional activities [18,19,20]. In the VAT, the subject is asked to perform active horizontal or vertical head movements over a rising frequency, ranging from 0.5 to 5.9 Hz, which can detect a closer-to-real state of the vestibular function [19]. Additionally, the VAT is much easier to perform since a computer controls it, induces less dizziness, and is better received by patients [21].

The reliability and repeatability of VAT have been tested in several studies. For example, Corvera et al. [22], in a drug trial, examined the efficacy of flunarizine in the treatment of acute vestibular neuritis by monitoring the VOR with the VAT. Their findings showed that the great variability in VAT results could reflect the variability in the patient’s responses to the treatment. Perez et al. [23] investigated the VAT results in patients with disabling MD and evaluated the change in VAT values immediately after intratympanically given gentamicin was stopped. They found that the inability of MD subjects to engage in high-frequency head movements was correlated to the intratympanic gentamicin treatment. However, VAT findings in the differential diagnosis of MD and VM patients are not yet fully known. Their role in the differential diagnosis of the two conditions warrants further research.

In this study, we primarily aimed to conduct an in-depth study on the differential diagnosis between MD and VM on the basis of different VAT parameters. Second, we further explored the clinical value of different frequencies with the horizontal gain in MD, VM and MD with migraine patients.

## 2. Materials and Methods

### 2.1. Patients and Inclusion/Exclusion Criteria

A retrospective study was performed at the Medical Center for Vertigo and Balance Disorders, Wuhan Union Hospital of Tongji Medical College, from January 2019 to January 2021. All subjects were entered into the study and this study has received approval from the Ethics Committee of the Wuhan Union Hospital (No. 20210873), Wuhan, China. 

The diagnostic criteria for MD involved two categories: definite MD and probable MD, and the criteria were established by the Classification Committee of the Barany Society [1]. The diagnostic criteria of definite VM and probable VM were formulated by the Committee for Classification of Vestibular Disorders of the Barany Society and the Migraine Classification Subcommittee of the International Headache Society (IHS) [5]. All patients’ medical records were reviewed by two specialists to decide if a patient met the diagnostic criteria of the definite or probable MD or VM for inclusion in the study. A subject was included as an MD with migraine patient if he or she satisfied the diagnostic criteria of MD (definite or probable MD) and VM (definite or probable VM) at the same time [8]. A subject was excluded if the patient (1) had a history of neck trauma and movement disorder, (2) had a severe hearing or visual impairment, (3) had mental disorders or serious physical diseases, (4) had severe systemic diseases and malignant tumors, (5) patients failing to complete VAT, (6) had undiagnosed vertigo, (7) age < 18 or pregnant women. 

After the elimination of ineligible candidates against the aforementioned criteria, eventually, 220 individuals were included in this study. All the data pertaining to these patients were collected, the medical records of these patients were reviewed, and information regarding their demography, clinical manifestations and auxiliary examinations (i.e., audiometric, vestibular function tests and radiographic examination) was retrieved, then transferred into EpiDate 3.1 database and recorded in a structured manner. 

### 2.2. Vestibular Autorotation Test (VAT)

Active head movements at high frequencies were examined by utilizing a VAT device and related software package (VATPLUS^®^) from WSR (Western System Research, Pasadena, CA, USA). Eye movements, horizontal and vertical, were observed by employing a pair of electro–oculographic electrodes put on each outer canthus and another pair of electrodes was placed above and below the left eye after skin cleaning. A band equipped with an electronystagmographic (ENG) amplifier and an angular velocity sensor was mounted on the head. During the test, the patient was required to fix their eyes on a target 120 cm away and start to move the head simultaneously with the beeps of the computer speaker. Patients were asked to perform head rotations on horizontal and vertical planes. Velocity was set at 0.5–0.9 Hz in the first 6 s, and it gradually rose from 1 to 6 Hz in the next 12 s. Then, gain, phase and asymmetry were recorded at the frequencies of 2.0, 2.3, 2.7, 3.1, 3.5, 4.3, 4.7 and 5.1 Hz. The test was performed three times on each plane. The coherence level was set at 0.6. Gain, phase and asymmetry values for eye movements on the horizontal plane and gain and phase for eye movements on the vertical plane were then computed. Gain, by definition, was the eye velocity amplitude divided by head velocity amplitude. Phase refers to the position of the eye (in degrees) in relation to that of the head. Asymmetry was defined as the deviation of the eyes from the middle line in horizontal head rotation. The retrospective VAT data of MD, VM and MD with migraine patients were collected from the non-ictal phase. The same well-trained technician carried out all the operations. The patient was sufficiently instructed, and the beeps from the computer speaker was generally loud enough to ensure that they could be heard by those suffering from hearing impairment. The VAT results (Appendix A) were reported as mean and SD of the horizontal gain, horizontal phase, asymmetry, vertical gain and vertical phase. In addition, the principle of VAT and the introduction of each parameter are shown in Appendix A.

### 2.3. Statistical Analysis

Data obtained in the study were analyzed statistically using the SPSS for Windows version 23.0 (IBM Corp.; Armonk, NY, USA) software. First, subjects were divided into MD, VM and MD with migraine groups. Continuous data are shown as mean ± SD (standard deviation) and categorical data as count (percentage). Continuous variables were evaluated using the Kruskal-Wallis one-way analysis of variance (non-parametric method). Categorical variables between the different groups were performed using the Chi-square test. Next, by using logistic regression, five VAT parameters, i.e., horizontal gain/phase, vertical gain/phase and asymmetry were analyzed in different groups. Finally, ROC curves were used for the analysis of the diagnostic accuracy, sensitivity, specificity, positive predictive value, and negative predictive value of VAT parameters in terms of the differential diagnosis of MD and VM groups. A *p*-value < 0.05 was considered statistically significant, *p*-value < 0.01 was very statistically significant, and *p* < 0.001 was extremely statistically significant. 

## 3. Results

### 3.1. Demographic Characteristics of Patients

A total of 220 patients were enrolled in this study, including 69 patients with MD (42 males and 27 females), with an average age of 50.22 ± 12.90 years. Seventy-nine patients had VM, including 30 males and 49 females, with an average age of 47.51 ± 14.15 years. There were 72 patients who had been diagnosed as having MD with migraine, including 20 males and 52 females, aged 42.44 ± 10.99 years. (Table 1).

### 3.2. VAT Parameters in VM and MD Patients

Comparison of the percentages of different VAT parameters between MD and VM are presented in Appendix A. By using univariate binary logistic regression, the VAT parameters in MD and VM groups were compared, and the result showed that the horizontal gain of VAT significantly outperformed other parameters in distinguishing MD and VM (*p* < 0.05) (Table 2). Multivariate binary logistic regression analysis of VAT parameters also exhibited that the horizontal gain could best distinguish between MD and VM (Appendix A). Due to the collinearity of vertical gain and phase, only one index could be used in multivariate binary logistic regression. The ROC curve of VAT parameters was drawn and analyzed in the same coordinate system to more intuitively observe the power of each parameter in distinguishing MD and VM (Appendix A). The sensitivity, specificity, positive predictive value, negative predictive value and accuracy of different VAT parameters in the differential diagnosis of between MD and VM patients are presented in Figure 1b. The sensitivity, specificity, positive predictive value, negative predictive value, and accuracy of the horizontal gain were 95.7%, 50.6%, 62.9%, 93% and 71.6%, respectively, in the differentiation between VM and MD (Figure 1b). In terms of horizontal gain, all the indicators outperformed the horizontal phase, vertical gain, vertical phase and asymmetry, though the specificity was relatively low. In addition, the ROC curve of VAT parameters was drawn and analyzed in the same coordinate system to more intuitively show the power of each parameter in distinguishing MD and VM. The AUCs of the ROC curves for horizontal gain, horizontal phase, vertical gain, vertical phase and asymmetry were 0.776, 0.572, 0.527, 0.527 and 0.529, respectively (Figure 1a). The cut-off values of the ROC curves of the horizontal gain, horizontal phase, vertical gain, vertical phase, and asymmetry were 0.320, 0.462, 0.526, 0.526 and 0.488, respectively (Appendix A). Therefore, among the five parameters of VAT, the horizontal gain was the greatest, indicating that it had the highest differential diagnostic power. 

### 3.3. Comparison of Different Frequencies of Horizontal Gain in MD, VM and MD with Migraine Patients

We further analyzed various frequencies of horizontal gain. The test value and standard value of horizontal gain were calculated for each frequency (2~6 Hz), and the proportion of horizontal gain decrease and increase was also computed for each frequency (Table 3). In most MD patients, the horizontal gain decreased in the range of 3~4 Hz, while in most of the VM patients, horizontal gain increased between the range of 2~3 Hz (Figure 2a,b). Moreover, more MD with migraine patients had an increased horizontal gain when the frequency was less than 5.0 Hz, and yielded a decreased horizontal gain when the frequency was greater than 5.0 Hz (Figure 2c).

## 4. Discussion

MD and VM are episodic vestibular syndromes defined by a set of associated symptoms such as tinnitus, hearing loss or migraine features during the attacks. They are both symptomatically diagnosed and share some features in terms of etiology, eliciting factors and clinical manifestations [6,24,25,26]. With the progression of the disease, they share increasingly more features, which renders it difficult to distinguish between them. However, there is no objective diagnostic method with specific results to establish a differential diagnosis between VM and MD and their diagnosis is based on medical history and symptoms. To our knowledge, few studies explored the clinical utility of vestibular autorotation tests in MD, VM and MD with migraine. Our study yielded three major findings. First, our study suggested the clinical utility of VAT, especially the horizontal gain, as an indicator, which may provide a sensitive and objective clue that helps distinguish between MD and VM. Second, in most MD patients, the horizontal gain decreased in the range of 3~4 Hz, while in most VM patients, horizontal gain increased between the range of 2–3 Hz. Third, most MD with migraine patients had an increased horizontal gain when the frequency was less than 5.0 Hz and had a decreased horizontal gain when the frequency was greater than 5.0 Hz.

It is well known that the caloric test and vHIT are important tests of vestibular functions and have shown varying results with both peripheral and central vestibular dysfunction. Yilmaz et al. [27] compared the results of the caloric test and vHIT of the VM and MD patient. The caloric test result showed that 66.1% of MD patients (*n* = 39) and 34% of VM patients (*n* = 17) had abnormal results, while abnormal gain of the lateral canal was found in 39% of their MD patients (*n* = 23) and 18% of VM patients (*n* = 9). The study concluded that MD patients display more abnormal results with both the caloric test and vHIT than VM patients. In another study by Hannigan et al. [28] showed that in 73 MD patients tested, 27 showed a dissociation between the caloric test (abnormal result) and vHIT (normal result), and, in stark contrast, of the 532 non-MD subjects, only 9 exhibited such dissociation. Similarly, a study by Pérez-Fernández et al. [29] examined the severity of endolymphatic hydrops in the cochlea and vestibule of patients with unilateral definite MD (*n* = 22) with respect to the results from the caloric test and vHIT; they found that there was a significantly higher degree of hydrops in the ear of patients with abnormal caloric test findings but normal vHIT results. They were led to conclude that endolymphatic hydrops severity in magnetic resonance imaging (MRI) evidences disparate vestibular test results [29,30]. Additionally, in gadolinium-contrasted MRI, previous studies confirmed that patients with MD showed endolymphatic hydrops (EH) in the vestibulum more frequently than in the cochlea, whereas EH was rare in patients with VM. Therefore, they concluded that inner ear imaging using gadolinium-contrasted MRI was also helpful in differentiating these two clinical disorders [31,32]. However, Eliezer et al. [33] thought that it could be difficult to distinguish between VM and MD at the early stage since EH on MRI is related to the degree of sensorineural hearing loss. The aforementioned findings showed that the results of vestibular function tests or an imaging examination varied and were not specific to MD and VM. Therefore, a more accurate diagnostic tool for distinguishing between the two conditions will be warranted.

In this study, we compared the VAT parameters of MD and VM patients and found that the horizontal gain of VAT significantly outdid other parameters in distinguishing between MD and VM. What is more, the sensitivity, specificity and accuracy of the horizontal gain was 95.7%, 50.6% and 71.6%, respectively, in the differentiation between VM and MD. The high sensitivity can incite the physician to dig thoroughly through the history, thus, decreasing the risk of missing a diagnosis of MD or VM. However, low specificity may cause false positives would lead to misdiagnosis. High sensitivity and low specificity are mainly attributed to two factors: (i) The detection frequency covers a wide range (2–6 Hz), therefore, the sensitivity is relatively high. (ii) The relatively low specificity can be ascribed to vestibular compensation, which leads to a subset of patients not having intrinsic pathology but yielding normal results due to their vestibular compensation during the detection. Low specificity is usually a manifestation of vestibular function recovery after compensation. 

However, there existed no significant difference in the value of the vertical test between the VM and MD groups, which might be ascribed to the short distance in the vertical direction of the eye when moving up and down during the vertical test [34]. This distance may not suffice to effect appreciable pathological findings, such as leading corrective saccades. In addition, the abnormal rate of the horizontal phase in VM and MD groups was similar. Given that the lesions in any pathway of VOR, including central and peripheral damage of the vestibule, can produce VAT phase delay [35]. The detection rate of asymmetrical abnormalities in MD group (26.1%) was higher than that in VM group (20.3%). This can be explained by the assumptions that VM was a non-organic lesion, and bilateral vestibular function was essentially symmetrical during the non-ictal phase, while, with MD, vestibular impairment on the lesion side was mostly unilateral, and, as a result, the detection rate of asymmetry was relatively high. Therefore, we propose the horizontal gain be used as a diagnostic tool for the differentiation between MD and VM since it may provide a more valuable clue.

VOR is one of the main balance-protection mechanisms and it stabilizes visual images on the retina to ensure unblurred vision during horizontal and vertical head movements [28,36,37]. The VAT is a high-frequency and active head rotation test used to subjectively evaluate the VOR and its function. Peripheral and central lesions of the vestibular nervous system can cause pathway dysfunction and may manifest characteristic frequency patterns on VAT parameters [38,39]. In the present study, we further explored the clinical value of different frequencies with horizontal gain in MD and VM patients. In most MD patients, we found that the horizontal gain decreased in the range of 3–4 Hz while in most of the VM patients, horizontal gain increased between the range of 2–3 Hz. Similar to our results, the findings by Thungavelu et al. [21] showed that VAT parameters in VM patients had elevated horizontal gain at a frequency of 2, 3, 4 and 5 Hz, a horizontal phase delay at a frequency of 2, 4, 5 and 6 Hz, had elevated vertical gain at the frequency 2–6 Hz and vertical phase delay at the frequency 4–6 Hz, as compared to healthy subjects. In addition, Gökgöz et al. [40] found that phase values of horizontal VOR at 2.0, 2.3 and 2.7 Hz were significantly higher in the decompensated MD patients than their compensated counterparts. The above-mentioned studies showed that vestibular functional impairment in patients was not all abnormal across the whole frequencies, but the functional impairment mainly involves certain frequencies, which is in line with the frequency characteristics of vestibular function.

When the peripheral structure of the vestibule is damaged and the primary reflex pathway is incomplete, VOR is impaired and the gain decreases [41]. If the central structure of the vestibule is abnormal and its inhibitory effect on the vestibular nucleus is weakened, VOR hyperfunction will result and the gain is increased [42,43]. However, in clinical practice, some patients satisfy the diagnostic criteria of both MD and VM and are diagnosed as having MD with migraine [7,9]. Our results suggested that more MD with migraine patients had an increased horizontal gain when the frequency was less than 5.0 Hz and a decreased horizontal gain when the frequency was greater than 5.0 Hz. A previous study [6] showed that, clinically, MD with migraine seemed to combine the features of both MD and VM, and was not a separate entity. Besides, MD with migraine may shift from MD predominance to VM predominance at different stages of the disease. Thus, it is difficult to use one single value of horizontal gain for identifying MD with migraine patients. It remains to be further studied whether MD with migraine is simply reflective of the partial symptom overlap of MD and VM, or MD with migraine involves some shared pathophysiologic processes. 

### Limitations

There were several limitations to this study. First, this was a retrospective study conducted in a tertiary care hospital, which might result in a selection bias towards patients with more severe forms of MD and VM. Second, retrospective VAT data of patients were collected during the non-ictal phase, as data are generally not available for the acute phase. As a result, we were not certain if the two different phases had the same results. Third, to help resolve our uncertainties, a larger prospective study should be carried out for MD, VM and MD with migraine patients in definite or probable symptomatic periods. 

## 5. Conclusions

Our study suggested that VAT, especially the horizontal gain, as an indicator, may provide a sensitive and objective clue that helps distinguish between MD and VM. In addition, given that MD with migraine seems to be a hybrid of MD and VM, detailed history inquiry and audio-vestibular examinations can be used in combination to achieve a more accurate differential diagnosis. Our results provided a more objective reference for clinicians in the diagnosis of MD, VM or MD with migraine. 

## Figures and Tables

**Figure 1 brainsci-12-01432-f001:**
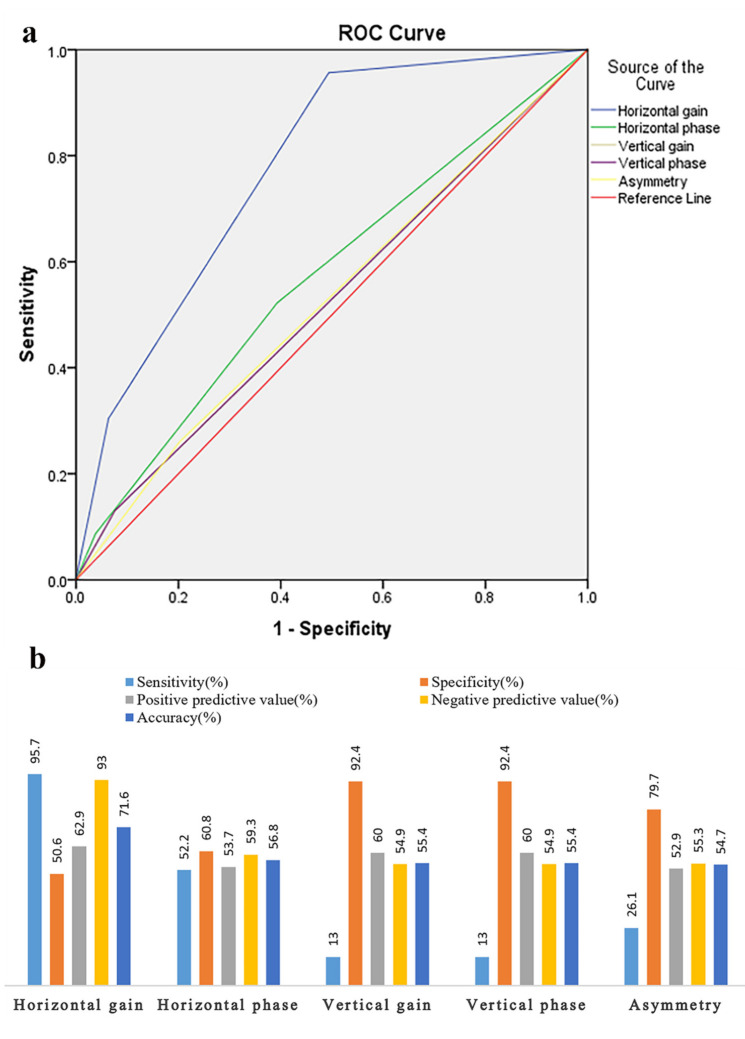
ROC curves of VAT in the study. (**a**) ROC curve analysis of different VAT parameters in the differential diagnosis of between MD and VM patients. Light blue line, horizontal gain; green line, horizontal phase; yellow line, vertical; purple line, vertical phase; red line, reference line. (**b**) The value of different VAT parameters in the differential diagnosis between MD and VM patients. MD, Menière’s disease; VM, vestibular migraine; VAT, vestibular autorotation test.

**Figure 2 brainsci-12-01432-f002:**
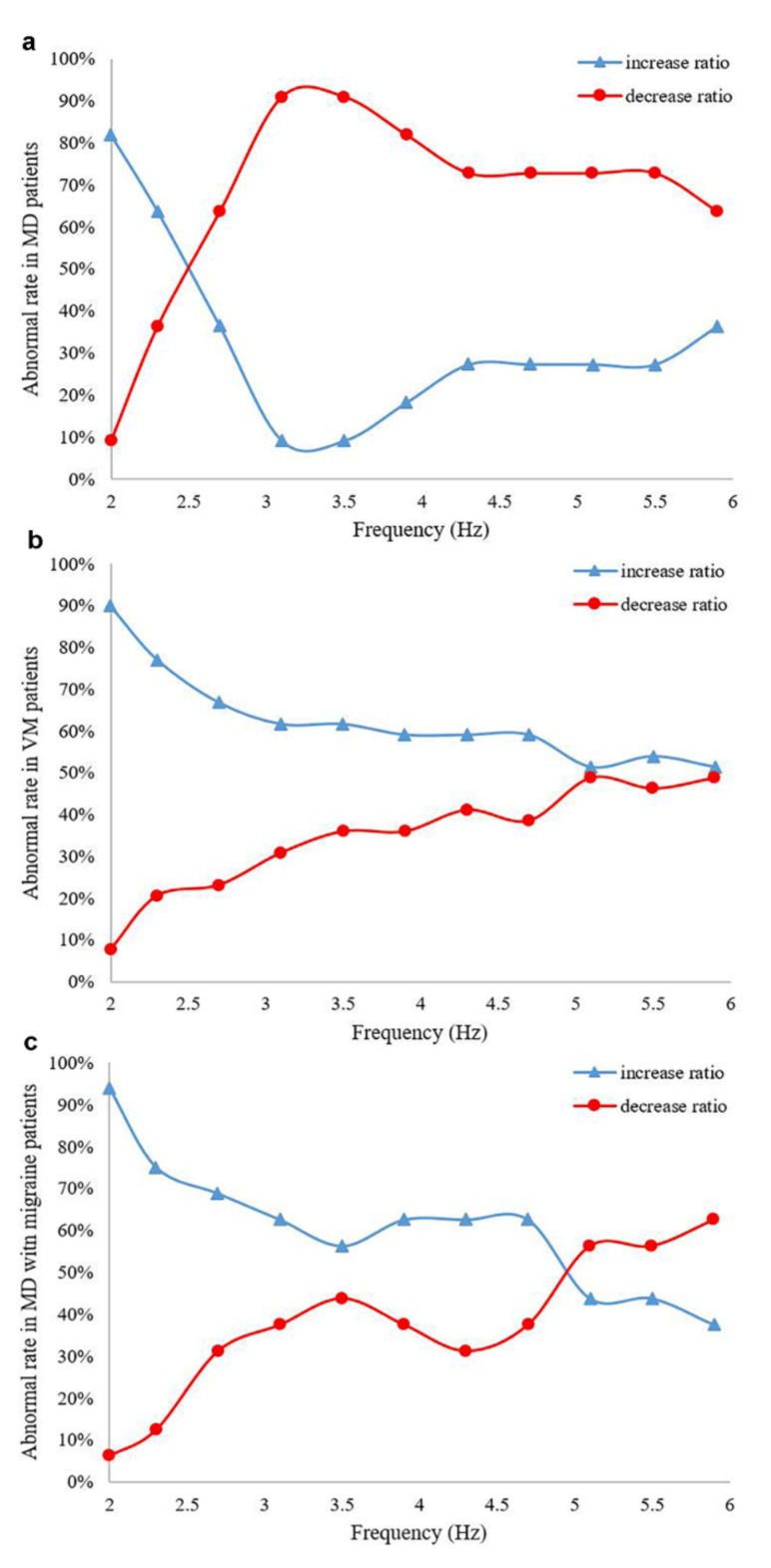
The proportion of the horizontal gain decrease in MD, VM and MD with migraine patients. (**a,b**) blue line, MD patients; light red line, VM patients; in most MD patients, the horizontal gain decreased in the range of 3–4 Hz, while in most of the VM patients, horizontal gain increased between the range of 2–3 Hz. MD, Menière’s disease; VM, vestibular migraine. (**c**) Blue line, MD with migraine patient’s horizontal gain increase ratio; light red line, MD with migraine patient’s horizontal gain decrease ratio. More MD with migraine patients had an increased horizontal gain when the frequency was less than 5.0 Hz and had a decreased horizontal gain when the frequency was greater than 5.0 Hz.

**Table 1 brainsci-12-01432-t001:** Demographic characteristics of patients in the three diagnostic groups.

	Disease			*p-*Value		
Variable	MD (*n* = 69)	VM (*n* = 79)	MD with Migraine (*n* = 72)	MD vs. VM	MD vs. MD with Migraine	VM vs. MD with Migraine
Age (range), year	50.22 ± 12.90	47.51 ± 14.15	42.44 ± 10.99	0.032 *	<0.001 ***	0.017 *
Gender (female)	27 (39.1%)	49 (62.0%)	52 (72.2%)	0.009 **	<0.001 ***	0.247
Illness duration	12 months	5 months	11 months	<0.001 ***	0.248	<0.001 ***
Vertigo/dizzy	67 (97.1%)	73 (92.4%)	67 (93.1%)	0.370	0.473	1.000
Visual motion	40 (58%)	50 (63.3%)	44 (61.1%)	0.622	0.835	0.914
Nausea and vomiting	55 (79.8%)	46 (58.2%)	48 (66.7%)	0.009 **	0.120	0.368
Hearing impairment	58 (84.1%)	30 (38.0%)	40 (55.6%)	<0.001 ***	<0.001 ***	0.045*
Tinnitus	62 (89.9%)	51 (64.6%)	56 (77.8%)	<0.001 ***	0.087	0.108
Aural fullness	50 (72.5%)	42 (53.2%)	45 (62.5%)	0.025 *	0.279	0.320
Light-photophobia	15 (21.7%)	62 (78.5%)	45 (62.5%)	<0.001 ***	<0.001 ***	0.048*
Sound-phonophobia	14 (20.3%)	59 (74.7%)	47 (65.3%)	<0.001 ***	<0.001 ***	0.278

Abbreviations: MD, Menière’s disease; VM, vestibular migraine. * *p* < 0.05, ** *p* < 0.01, *** *p* < 0.001.

**Table 2 brainsci-12-01432-t002:** Analysis of different VAT parameters to differentiate MD and VM patients by univariate binary logistic regression.

Parameter	Horizontal Gain			Horizontal Phase			Vertical Gain			Vertical Phase			Asymmetry	
Classification	Decrease	Normal	Increase	Decrease	Normal	Increase	Decrease	Normal	Increase	Decrease	Normal	Increase	Normal	Abnormal
OR	1.00	0.32	0.02	1.00	0.54	0.34	1.00	0.55	1.00	1.00	0.55	1.00	1.00	1.39
95%CI	-	0.10~0.92	0.00~0.08	-	0.12~2.35	0.08~1.47	-	0.14~2.03	0.11~8.94	-	0.14~2.03	0.11~8.94	-	0.64~2.99
*p*	<0.001 ***	0.03 *	<0.001 ***	0.21	0.41	0.15	0.56	0.37	1.00	0.56	0.37	1.00	0.23	0.40

Note: * *p* < 0.05, *** *p* < 0.001; Abbreviations: MD, Menière’s disease; VM, vestibular migraine; VAT, vestibular autorotation test.

**Table 3 brainsci-12-01432-t003:** Abnormal rate in different frequencies of the horizontal gain in the MD, VM and MD with migraine patients.

Frequency (Hz)	MD (*n* = 60) (%)		VM (*n* = 79) (%)		MD with Migraine (*n* = 72) (%)	
	Decrease	Increase	Decrease	Increase	Decrease	Increase
2	9.1	81.8	7.7	89.7	6.3	93.8
2.3	36.4	63.6	20.5	76.9	12.5	75.0
2.7	63.6	36.4	23.1	66.7	31.3	68.8
3.1	90.9	9.1	30.8	61.5	37.5	62.5
3.5	90.9	9.1	35.9	61.5	43.8	56.3
3.9	81.8	18.2	35.9	59.0	37.5	62.5
4.3	72.7	27.3	41.0	59.0	31.3	62.5
4.7	72.7	27.3	38.5	59.0	37.5	62.5
5.1	72.7	27.3	48.7	51.3	56.3	43.8
5.5	72.7	27.3	46.2	53.8	56.3	43.8
5.9	63.6	36.4	48.7	51.3	62.5	37.5

Abbreviations: MD, Menière’s disease; VM, vestibular migraine.

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
