# Peer review of "Diagnostic Value of the Vestibular Autorotation Test in Menière’s Disease, Vestibular Migraine and Menière’s Disease with Migraine"

_brainsci, 2022, doi:10.3390/brainsci12111432_

Round 1

Reviewer 1 Report

The authors report a retrospective analysis of 69 Meniere’s disease (MD) patients, 79 vestibular migraine (VM) patients, and 72 patients with both MD and VM (MDVM). Clinical diagnoses were done according the diagnostic criteria of The Barany Society. All patients underwent vestibular autorotation test (VAT). Sensitivity, specificity and accuracy of the horizontal gain were 95.7%, 50.6% 23 and 71.6%, respectively, for the differentiation between VM and MD. In most MD patients, the horizontal gain decreased in the range of 3-4Hz while in most of VM patients, horizontal gain increased 25 in the range between 2-3Hz. The authors suggest that the horizontal gain of the VAT may be helpfull in differential diagnosis between MD and VM.

This is an interesting study. I only have some minor suggestions:

The reader of Brain Science will not necessarily be familiar with VAT, so I suggest to include a figure in the Methods section to show how this examination works, what the patient has to do and what parameters are measured.

Specificity was rather low. Thus, it has to be discussed what may be the reasons and what are the consequences when considering it as a test for differentiation between MD and VM.

I think that modern MRI sequences are able to image the endolymph hydrops, so that it suggests itself to use MRI of the inner ear to get additional clues for the differentiation between VM and MD. That could be discussed.

I wonder if the patients also got calorics and/or vHIT? It could also be interesting to compare these results to the VAT results in the patient groups.

Author Response

Please see the attachment,Thank you.

Reviewer 2 Report

Line 107:

The authors stated: “or family history”, they should define family history of what? MD, VM or both or other autoimmune disease? Why do they exclude family history of MD in MD? Or of VM in their VM patients? This needs clarification.

Line 272-275 and 229-234:

The authors stated in the limitations: “the patients were not sorted out into VM or MD in inter-ictal phase and acute stage. As a result, we were not certain about if the two different phases had the same results”.

While they previously stated in the discussion Lines 229-234: “. The detection rate of asymmetrical abnormalities in MD group (26.1%) was higher than that in VM group (20.3%). This can be explained by the assumptions that VM was a non-organic lesion, and bilateral vestibular function was essentially symmetrical during the non-ictal phase, while, with MD, vestibular impairment on the lesion side was mostly unilateral, and, as a result, the detection rate of asymmetry was relatively high.  

The authors did not confirm the test retrospective data were taken from a test that has been done in the non-ictal phase. Furthermore, it was not stated whether these percentages were or not statistically significant between MD and VM in Table S1? This should be statistically dealt with before interpreting the results completely to get accurate assumptions.

Line 121-122:

In the methodology, the authors stated: “During the test, the patient was required to fix the eyes on a target 120 cm away and started to move the head simultaneously with the beeps of the computer speaker”. And the authors stated in table 1: “Hearing impairment was in58 (84.1%) of MD  30 (38.0%) of VM”, so were the beeps from the computer speaker loud enough to ensure being heard by those having hearing impairment to assure accurate technique? This should be stated.

Table 1

The authors stated the Key: “*p<0.05, **p<0.01, ***p<0.001”, however, there is no asterisks seen in the table. These should be clearly shown. Furthermore, they stated in the statistical analyses: “A p-value less than 0.05 was considered to be statistically significant”, without mentioning p<0.01, or p<0.001.

Table 2 and Supplemental Table S2:

The authors stated the “Vertical gain/phase” together as one unit! Even if the values were similar? Why not statistically dealt with separately like the Horizontal gain, and Horizontal /phase?

figure 1:

It was shown from the figure that the Horizontal gain had the highest diagnostic accuracy, sensitivity, specificity, positive predictive value, negative predictive value of all the VAT parameters in terms of the differential diagnosis of MD and VM groups. This needs to be stated clearly also in the text.  They only stated:” Fig. 1b. The sensitivity, specificity and accuracy of the horizontal gain was 95.7%, 50.6% and 71.6%, respectively, in the differentiation between VM and MD (Fig. 1b).” without mentioning that they were the highest or best values.

figure 2b:

In figure 2a:   only one curve for MD and only one curve for VM values across frequencies. But in figure 2b:  unclear why 2 curves for MD/VM,  while they are one group?

Supplemental Figures:

Figures A1 and A2: Text is not clearly seen; figures should be enlarged to make text easy to read.

Author Response

Please see the attachment,Thank you.

Reviewer 3 Report

Authors evaluated the usefulness of vestibular autorotation test (VAT) in differentiating MD, VM and concurrent MD/VM. Comparing VAT parameters in patients with MD and VM has important clinical values. 

In figure 1, ROC curves are demonstrated in each parameter, and the most appropriate cut-off values seemed to be used to classify “decrease”, “normal” or “increase” in table S1. Authors need to show cut-off values in detail to bridge the gap between figure 1 and table S1.

Author Response

Please see the attachment, thank you
